# Prediction of Protein–Ligand Interaction Based on the Positional Similarity Scores Derived from Amino Acid Sequences

**DOI:** 10.3390/ijms21010024

**Published:** 2019-12-18

**Authors:** Dmitry Karasev, Boris Sobolev, Alexey Lagunin, Dmitry Filimonov, Vladimir Poroikov

**Affiliations:** 1Department of Bioinformatics, Institute of Biomedical Chemistry, Moscow 119121, Russia; boris.sobolev@ibmc.msk.ru (B.S.); alexey.lagunin@ibmc.msk.ru (A.L.); dmitry.alekseevich.filimonov@gmail.com (D.F.); vladimir.poroikov@ibmc.msk.ru (V.P.); 2Department of Bioinformatics, Russian National Research Medical University, Moscow 117997, Russia

**Keywords:** protein–ligand interaction, local sequence similarity, prediction of protein targets

## Abstract

The affinity of different drug-like ligands to multiple protein targets reflects general chemical–biological interactions. Computational methods estimating such interactions analyze the available information about the structure of the targets, ligands, or both. Prediction of protein–ligand interactions based on pairwise sequence alignment provides reasonable accuracy if the ligands’ specificity well coincides with the phylogenic taxonomy of the proteins. Methods using multiple alignment require an accurate match of functionally significant residues. Such conditions may not be met in the case of diverged protein families. To overcome these limitations, we propose an approach based on the analysis of local sequence similarity within the set of analyzed proteins. The positional scores, calculated by sequence fragment comparisons, are used as input data for the Bayesian classifier. Our approach provides a prediction accuracy comparable or exceeding those of other methods. It was demonstrated on the popular Gold Standard test sets, presenting different sequence heterogeneity and varying from the group, including different protein families to the more specific groups. A reasonable prediction accuracy was also found for protein kinases, displaying weak relationships between sequence phylogeny and inhibitor specificity. Thus, our method can be applied to the broad area of protein–ligand interactions.

## 1. Introduction

Recognition of specific interactions of biological macromolecules is necessary for the study of regulatory processes in biological systems. The identified protein ligands can be applied for the study of signaling and metabolic pathways. Finding the novel target for small-molecule compounds is the important step in drug design. Machine learning methods provide a relatively fast prediction of the target for known drugs and vice versa [1]. Investigators apply various approaches, which can be distinguished by the underlying mathematical techniques and training data [2]. However, the primary aim of a prediction tool is its ability to provide reliable results for the specific applicability domain. Indeed, the prediction power mainly depends on the type of input data and the method of their description. The training sets are commonly classified based on the ligand affinity value to a protein target. However, they can represent differed data types, such as the chemical structure of small-molecule ligands, various data on protein targets, etc.

(Q)SAR (Quantitative Structure-Activity Relationship) methods build models, in which the protein identifiers are class-forming features without considering protein features [3,4]. In this case, the prediction of a new target (not covered by the training data) is impossible. It should be noticed that only 11% of the human proteome had established ligands [5]. The highly accurate methods for prediction of protein targets can reduce the labor and financial costs of the design and discovery of new pharmaceutical agents [6]. Proteochemometric approaches solve the mentioned task by involvement of the information on proteins [6,7,8,9,10,11,12,13,14,15,16,17,18,19,20,21,22,23,24,25].

Amino acid sequences, as well as characteristics calculated by their analysis, are the most representative data and widely used for the primary search of potential drug targets.

Several tools predicting the protein targets for ligands use the protein similarity matrices obtained for entire protein sequences by pairwise alignment [8,12,16,19,21,23,25]. Such a method can recognize targets if the ligand specificity well correlates with overall sequence proximity, detected by phylogenic studies. However, the specificity of drug compounds may not coincide with phylogenic relations [26]. Several publications display the cases when relatively few residues determine the protein functional specificity within the family of homologs [27,28,29,30,31,32,33,34]. The protein kinase family demonstrates a vivid example of this situation [35].

Several methods derive the protein descriptors from the multiple sequence alignment [22,36]. However, these techniques do not always provide an accurate match of sequence positions, which can present a few functionally significant residues. The integrated score of sequence similarity, such as the identity percentage, can hide the impact of the individual amino acid positions on the ligand specificity determination.

We developed a method based on the local similarity of amino acid sequences. The sequence fragments of the given length (the *F* parameter) from the query protein sequence and training protein sequences are compared. By this way, each position of the query sequence gets the score. These values are input data to the classifier, which estimates the protein specificity to the ligands. Earlier, we have demonstrated that this tool can be applied for functional annotation of proteins [37] and identification of functionally important residues in diverged paralog proteins [34].

In this study, our algorithm was tested on the enzyme set representing different families (and therefore various fold types), as well as datasets related to the individual families. We handled the full-size sequences and sequenced parts limited by the domain boundaries. We demonstrated that the suggested approach is applicable for protein data with a significant degree of heterogeneity, unlike the many existing methods often fitted to specific studied areas [38,39].

## 2. Results and Discussion

We evaluated the performance of our approach with the length of the compared sequence regions (*F* parameter) equal to 7 or 30 (see Materials and Methods, Section 3.2—Positional Similarity Scores).

### 2.1. Evaluation on Gold Standard and PASS Targets Datasets

Evaluation of our method on all Gold Standard datasets, including GPCR (G-Protein Coupled Receptors), Ion Channels, and Nuclear Receptors and Enzyme, brought highly accurate results, especially at *F* = 30. The AUC (Area Under Curve) values calculated by ROC (Receiver Operating Characteristic) and obtained with the Leave-One-Out Cross-Validation procedure are presented in Table 1 and Figure 1.

Application of the PASS Targets GPCR dataset brought highly accurate results, like those obtained for the Gold Standard set. A slight decrease in AUC values in the case of the PASS Target GPCR dataset could be due to a more considerable diversity of the training data (the highest number of ligands). For both sets on Nuclear Receptors the most accurate estimation is obtained.

Interestingly, testing on the Gold Standard dataset for Ion Channels showed the most significant difference in an accuracy at different *F* values, revealing the high AUC at *F* = 30 and lower AUC at *F* = 7. This result suggests that the distant interactions of amino acid residues are essential for ligand specificity determination. The data on Ion Channels extracted from PASS Targets were too poor, likely influencing the relatively low AUC values.

The set Enzyme of Gold Standard consists of proteins from various families. For this reason, different classes of ligand specificity have to relate to the particular protein families, which present the non-overlapped groups of homological sequences. The high accuracy of prediction was somewhat due to the overall sequence similarity. So, each of the datasets on GPCR, Ion Channels, and Nuclear Receptors seems to be partitioned into groups of close homologs well coincided with the ligand specificity classes. This circumstance explains the successful testing of the Gold Standard with several approaches [12,16,19,21,23,25,41].

### 2.2. Protein Kinases

The more sophisticated task is related to the cases when ligand specificity is not strongly correlated with the overall sequence similarity.

The datasets retrieved from Karaman and coworkers [42] provided the interactions of kinase domains with inhibitors, allowing for the restriction of the studied area by the residues involved in the interaction with ligands. We obtained the moderate accuracy estimation, but the prediction with the stronger threshold (Kd less than 0.1 μM) brought a higher AUC at both *F* values. The highest AUC values were obtained at *F* = 30, though the difference related to the *F* value was not dramatic (Figure 2 and Table 2). However, the distant inter-positional interactions within the protein domain could influence the ligand binding. The prediction on the dataset from Gao and coworkers [43] displayed less accuracy (Table 2).

The moderate accuracy detected on protein kinase data seems to be related to the structural features of kinase-inhibitor binding. The studied ligands are known to interact with kinase at the ATP-binding cleft, which is structurally conserved in proteins of a given family [35].

So, the substitution of single residues can modulate the ability of ligand binding that coincides with our prediction of the ligand-specific positions in protein kinase sequences [34]. These observations suggest a there is a weak correlation between the inhibitor specificity spectrum and phylogeny. You can see such a case in Figure 3: The same ligand reveals affinity to the close as well as remote homologs.

Our method allowed taking into account the impact of separate regions and single residues. The approach used resulted in a reasonable accuracy of ligand specificity of protein kinases, despite the peculiarities mentioned.

### 2.3. Comparing with Other Methods

We compared the performance of our approach with the other methods, tested on the same datasets retrieved from Gold Standard [12,16,19,21,23,25,41]. These results were compiled by Shi and coworkers [41]. In all studies, the accuracy estimations were given as AUC values obtained by ROC analysis. As can be seen from Table 3, our approach showed a better result than most other methods. In particular, some authors obtained an AUC less than 0.7 for Nuclear Receptors, while our approach brought the highest AUC (0.99) for this dataset. The AUC values calculated for GPCR were less than 0.9 in three cases versus 0.94 in our results. The best estimates for Enzymes and Ion Channels sets have been obtained by all methods considered.

## 3. Materials and Methods 

### 3.1. Training Sets

We tested our method on several training sets partitioned into classes of small-molecule ligand specificity. Thus, proteins binding the given ligand were composed of a specificity class, while others formed the complementary class (or class complement). For achieving reliable classification results, each ligand specificity class should contain at least five proteins. Application of such a restriction provided the most reliable results according to our earlier studies [34]. All sequences were extracted from UniProt KnowledgeBase (https://www.uniprot.org/). The characteristics of the datasets used are shown in Table 4.

The Gold Standard dataset collected by Yamanishi and co-authors [8] from several databases (KEGG BRITE, Brenda, SuperTarget, and DrugBank) includes four sets presenting human proteins. The Enzymes set contains the enzymes related to different families. The three specific sets include data on GPCR (G-protein Coupled Receptors), Nuclear Receptors, and Ion Channels. The sequences without evidence of binding (conditionally negative examples) formed the complement of each ligand-specificity class. Thus, the affinity to a given ligand was presented using a binary classification (interacts, does not interact). Studying the Gold Standard sets, we used the full-size sequences with the corresponding UniProt identifiers.

Several research groups used the Gold Standard for testing their programs [8,12,16,19,21,23,25,41]. It allowed us to compare our method with the other approaches. However, these datasets are rather old, and we decided to test our algorithm additionally using data recently prepared for the (Q)SAR-based prediction of small-molecule ligands at the web-service PASS Targets (http://www.way2drug.com/passtargets/) [40]. We extracted data on the interaction with ligands of four promising drug target classes: protein kinases, GPCR, Nuclear Receptors, and Ion Channels. In this case, we also studied the full-size UniProt sequences. The classes were divided into positive and conditionally negative examples [40].

We also used data on protein kinases and their inhibitors presented by two experimental studies [42,43]. The protein kinases are the intensively studied drug targets considered as promising anticancer drug targets [44,45]. In this case, the complement class included truly negative examples. Both studies showed the results obtained by the particular team of authors under the same conditions. Karaman and co-authors [42] reported the data on multiple human kinases tested vs. numerous inhibitors. Establishing the cutoffs of constant dissociation (Kd), we referred the proteins displayed fewer values to a class, and the rest to a complement. Thus, two cutoff values 0.1 μM and 1.0 μM provided two sets (Table 4). As the authors worked with recombinant constructions containing kinase domains, we also used the sequences limited by domain boundaries.

Gao and co-authors [43] evaluated the interactions of multiple human kinases with various inhibitors. Based on the residual activity, the authors defined proteins as highly active, active, weakly active, and inactive related to each ligand. The active and highly active enzymes were defined as belonging to the class, while the others were belonging to the complement. The studied sequence cloned in recombinant constructions always covered the kinase domain and captured the more extensive areas in some cases. We used the sequences limited by the boundaries specified in the article [43].

### 3.2. Positional Similarity Scores

To describe the protein under study, each amino acid residue was estimated by the score calculated in terms of its surrounding in a sequence. Unlike the sequence alignment, our approach used the comparison of all sequence segments, providing to overcome the problems related with ambiguously aligned regions [34].

At the first step, the sequence of a test protein (*Q*) was compared with each training sequence (*K*). The amino acid positions (*n* = 1 … *Sequence Length*) of *Q* get the rates (*R*) obtained in the series of successive shifts between sequences *Q* and *K*.

With a certain shift value *h* between *Q* and *K*, the *R* values were calculated for all matched regions of a given length *F* within the overlapped area of compared sequences.
(1)$$R_{ih}=\sum_{j=i}^{j+F-1}sim\left(q_{j,}k_{j+h}\right),$$
where *i* is the position of the test sequence *Q*, and *sim*(*q_n_*, *k_m_*) is a similarity of matched residues according to an established measure (identity, Blosum62, or other substitution matrixes).

Thus, each *R_ih_* value is calculated as the summarized similarity of compared sequence regions.

Finally, each position *p* takes the score equal to the maximum of *R* values, obtained for all regions including *p* at all permitted shifts.
(2)$$S_{p}=\underset{h,i}{\max}\;R_{ih},p-F\leq i<p.$$

In this study, we used the identity of amino acid sequences as a similarity measure. The *F* values were set to 7 and 30 for the following reasons: Based on our preliminary studies, we suggested that the less fragment length is preferable for estimating the impact of the individual amino acid positions [34]. The larger *F* value better allowed accounting for distant inter-positional dependencies [36].

The positional scores are input data for the classifier predicting the protein belonging to the classes of ligand specificity.

### 3.3. Prediction Algorithm

For the prediction of protein functional specificity (including binding to ligands), we applied the modification of the PASS algorithm, which showed the effective estimation of the biological activities of chemical compounds [3]. The method is based on the naïve Bayesian classifier and uses the descriptors presenting the objects analyzed. For each *k*th training sequence, weight coefficients *a_k_* and *b_k_* define its belonging to a given class *C* and its complements, respectively. In this study, we used binary weights adopting the discrete values of 0 or 1.

The positional scores were obtained at the pairwise comparisons of a test sequence with *n* training sequences. The integrated score for each position *p* is found as follows:(3)$$t_{p}=\frac{\sum_{k=1}^{n}S_{pk}\times \left[a_{k}(C)-b_{k}(C)\right]}{\sum_{k=1}^{n}S_{pk}\times \left[a_{k}(C)+b_{k}(C)\right]},$$
where *S_pk_* is a similarity score of the position *p*, obtained by comparison with the *k*th sequence. The positional score values *t_p_* are averaged over all *m* positions of the test sequence as
(4)$$t=\sin\left[\frac{1}{m}\sum_{p=1}^{m}\arcsin(t_{p})\right].$$

The a priori score value *t*_0_ is calculated for the class and its complement as
(5)$$t_{0}=\frac{\sum_{k=1}^{n}a_{k}(C)-b_{k}(C)}{\sum_{k=1}^{n}a_{k}(C)+b_{k}(C)}.$$

Finally, the test sequence belonging to class *C* is estimated as follows:(6)$$B(C)=\frac{t-t_{0}}{1-tt_{0}}.$$

The statistic *B*(*C*) obtained varies from −1 (belonging to a complement) to +1 (belonging to a class). The value close to 0 shows the indefinite result.

Earlier, we have successfully applied the approach described for functional annotation of proteins [37,46] and determination of amino acid residues responsible for ligand binding [34].

### 3.4. Leave-One-Out Cross-Validation (LOOCV) Procedure

For estimating the predictive performance for the given ligand specificity class, the sequence set is divided into the class and the complement. At each step, one of the sequences is removed from the dataset and treated as a test protein. All results, obtained for all test sequences and all classes, were used for ROC analysis to estimate the prediction efficiency by area-under-curve (AUC) values.

### 3.5. Computation Time

For each of the datasets, the local similarity of each sequence to all the others was calculated. This number is 580 × 581 for the Gold Standard Enzymes dataset. Thus, for all datasets, more than 657,398 local similarity calculations were performed using a Hewlett-Packard HP X210 Workstation X64-based PC Intel Xeon CPU E31239 3.20 GHz, 8 GB memory, for about 1.5 h, or less than 10 milliseconds per pair of amino acid sequences.

## 4. Conclusions

In this work, we showed that our approach was suited for the prediction of protein–ligand interactions. Earlier, the method had proved its efficacy for functional annotation of the proteins, as well as the identification of functionally important amino acid residues. The technique allows predicting new targets for ligands with the already-known target spectrum. Efficiency evaluated on the standard training sets was not inferior to other published approaches. Moreover, our method showed efficiency in the case of ligand specificity groups, belonging to the same family of diverged proteins.

Despite the power of those methods developed by other authors, they often use a large number of descriptors and nontrivial mathematical calculations [6,7,8,9,10,11,12,13,14,15,16,17,18,19,20,21,22,23,24,25]. Our approach has advantages in the technology of assessment: Calculations are quite simple and do not require a lot of calculations, and we do not use complicated amino acid descriptors (any physicochemical descriptors or substitution matrixes).

We reach a high prediction accuracy with datasets from the well-known resource Gold Standard. Compared to other methods, our program showed better or comparable results. Other methods estimate the test and training sequences by pairwise alignment. Our method seems to be appropriate in case of a weak correlation between the ligand specificity and phylogeny. As the specificity of ligand binding can be related to a small number of residues [31,34], estimation of impact of the individual residues is better suited in the case of the diverged protein family. This situation is typical and can be illustrated by the example of protein kinases. We compiled and verified the sets derived from individual experimental studies on kinase-inhibitor interactions. This study demonstrated reasonable accuracy being tested on diverse data; thus, our approach significantly extends the applicability domain in proteochemometrics.

## Figures and Tables

**Figure 1 ijms-21-00024-f001:**
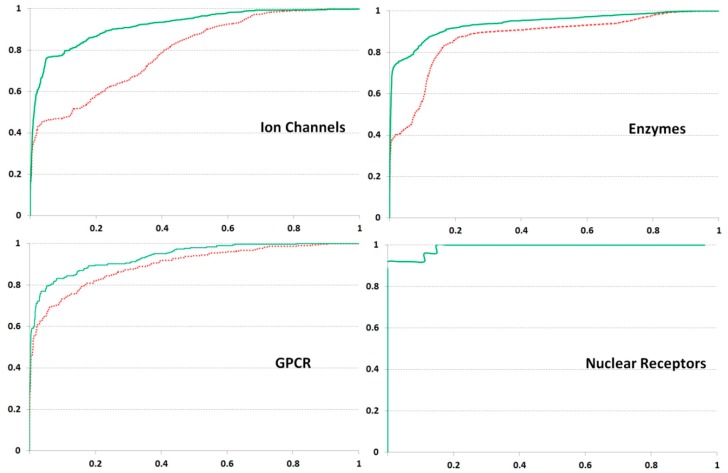
Results of ROC (Receiver Operating Characteristic) analysis obtained on Gold Standard datasets. *x*- and *y*-axes are coincided with False Positive and True Positive Rates, respectively. The green solid and red dashed lines depict the results obtained at *F* values 30 and 7, respectively. The results for Nuclear Receptors are displayed by the single line as practically identical.

**Figure 2 ijms-21-00024-f002:**
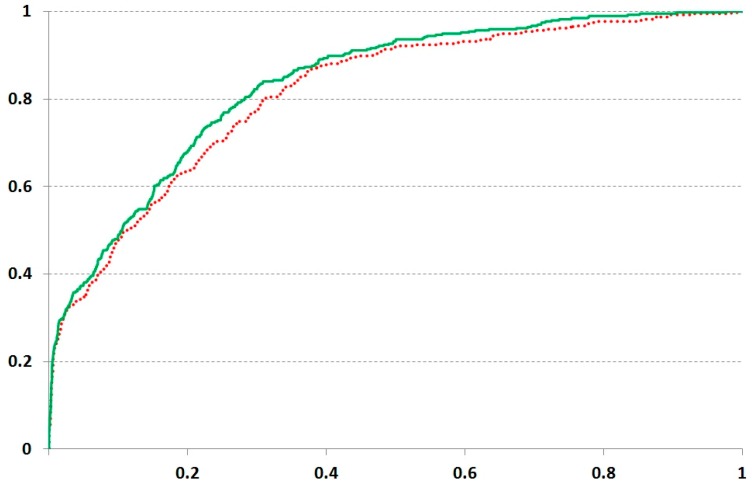
Results of ROC analysis obtained on the protein kinase set from Karaman and coworkers [42] at the 0.1 µM cutoff. The solid and dashed lines depict the results obtained at *F* values 30 and 7, respectively.

**Figure 3 ijms-21-00024-f003:**
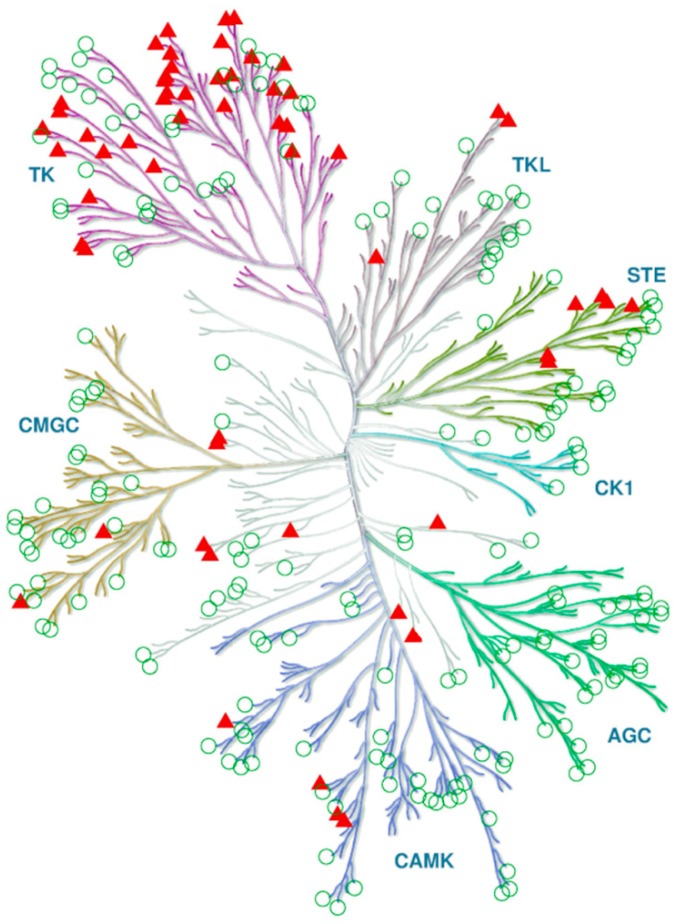
Protein targets of an inhibitor (VX-680) on the phylogenetic tree of human protein kinases. Triangles and circles depict the inhibited and not inhibited kinases, respectively. The data on inhibition at the 1.0 µM threshold was retrieved from [42]. The tree is reproduced from Cell Signaling Technology Inc. (www.cellsignal.com).

**Table 1 ijms-21-00024-t001:** Evaluation of our approach on the datasets received from Gold Standard and PASS Targets [40]. AUC (Area Under Curve) amounts were calculated at two *F* values.

Data Source	Dataset	*F* Parameter	
		7	30
Gold Standard [8]	Ion Channel	0.796	0.921
	GPCR	0.897	0.939
	Nuclear receptors	0.989	0.991
	Enzymes	0.874	0.941
PASS Targets [40]	GPCR	0.879	0.902
	Ion Channel	0.729	0.81
	Nuclear receptors	0.963	0.966

**Table 2 ijms-21-00024-t002:** AUC values obtained in testing our approach on protein kinases.

Dataset	*F* Parameter	
	7	30
Karaman et al., 2008 [8], K_d_ < 0.1 μM	0.815	0.835
Karaman et al., 2008 [8], K_d_ < 1.0 μM	0.790	0.801
PASS Targets dataset [40]	0.694	0.779
Gao et al., 2013 [43]	0.753	0.765

**Table 3 ijms-21-00024-t003:** The AUC values calculated by our approach and other methods on the Gold Standard datasets.

Dataset	Method							
	NetLapRLS [12]	WNN-GIP [19]	RLScore [23]	KBMF2K [16]	CMF [21]	NRLMF [25]	TMF [41]	Our Approach *
Enzymes	0.905	0.947	0.931	0.876	0.915	0.966	0.976	0.941
Ion Channels	0.914	0.950	0.937	0.938	0.905	0.964	0.972	0.921
GPCR	0.770	0.926	0.853	0.882	0.837	0.930	0.959	0.939
Nuclear Receptors	0.655	0.935	0.736	0.668	0.680	0.851	0.929	0.991

* The better results for each dataset are presented.

**Table 4 ijms-21-00024-t004:** The numbers of sequences and ligands used for testing of our method.

Data Sources	Target Type	Sequences	Ligands
Gold Standard [8]	Enzymes	581	97
	Ion channels	201	90
	GPCR	59	35
	Nuclear receptors	13	4
PASS Targets dataset [40]	Protein kinases	374	527
	Ion channels	14	3
	GPCR	76	197
	Nuclear receptors	15	14
Karaman M., 0.1 μM [8]	Protein kinases	220	26
Karaman M., 1 μM [8]	Protein kinases	261	30
Gao Y. [43]	Protein kinases	123	74

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
