# Peer review of "Prediction of Protein–Ligand Interaction Based on the Positional Similarity Scores Derived from Amino Acid Sequences"

_ijms, 2019, doi:10.3390/ijms21010024_

Round 1

Reviewer 1 Report

In their study, Karasev et al. present a methodology in order to predict the possible interaction of a ligand with a familiy of proteins. They have used several datasets to test their methods.

In my opinion, this paper is technical and does not bring so much novelty in the field (or maybe not enough emphasized). Thus, it is  not made for the audience of IJMS and has to be published in a more specialized journal. Furthermore,the paper needs to be improve on a methodological point of view (sequence alignment methods not described, Table 4 not currently describing the datasets, what is the meaning of the parameter F and why those values of 7 and 30, etc...). No clues about computation time have been given in the results. Consequently, for example, table 3 shows that TMF method is better on 3 cases than the methodology described in the paper. There is just a simple sentence in the conclusion, not supported by any part of the article.

Details have to be given to better explain the interest of this new methodology and what does it brings compared to the other one.

Author Response

Dear Reviewer,

We would like to express our gratitude for all the valuable comments and suggestions.

We have tried to do our best to improve our manuscript. Our responses are given below with the new sentences included in text of the manuscript. The modified text of the article in the attached file. The corrections are highlighted by yellow.

Yours sincerely,

On behalf of the co-authors:

Dmitry Karasev

Point 1. “In my opinion, this paper is technical and does not bring so much novelty in the field (or maybe not enough emphasized). Thus, it is not made for the audience of IJMS and has to be published in a more specialized journal.”

Response 1. We emphasized the items, which can be interesting for audience of the journal (lines 28-32): “Recognition of specific interaction of biological macromolecules is necessary for the study of regulatory processes in the biological systems. The identified protein ligands can be applied for study of signaling and metabolic pathways. Finding the novel target for small-molecule compounds is the important step in drug design. The machine learning methods provide the relatively fast prediction of the target for known drug and vice versa.”

Point 2. “Furthermore, the paper needs to be improve on a methodological point of view (sequence alignment methods not described, Table 4 not currently describing the datasets, what is the meaning of the parameter F and why those values of 7 and 30, etc.)”.

Response 2.

The section “4.2 Positional similarity scores” is rewritten and clarified (lines 203-223).

Table 4 (lines 167-168) was filled by the corresponding values.

We added the new paragraph into the “Introduction” (lines 60-65), which briefly describes our approach including the meaning of the F parameter: “We developed the method based on the local similarity of amino acid sequences. The sequence fragments of the given length (the F parameter) from the query protein sequence and training protein sequences are compared. By this way, each position of the query sequence gets the score. These values are input data to the classifier, which estimates the protein specificity to the ligands. Earlier, we have demonstrated that this tool can be applied for functional annotation of proteins [37] and identification of functionally important residues in diverged paralog proteins [34].”

The choice of the F values is explained in a new paragraph in “4.2 Positional similarity scores” (lines 219-223): “The F values were set to 7 and 30 for the following reasons. Based on our preliminary studies, we suggested that the less fragment length is preferable for estimating the impact of the individual amino acid positions [34]. The larger length is better allowed accounting for distant inter-positional dependencies [36].”

Point 3. “No clues about computation time have been given in the results.”

Response 3. We added section “4.5 Computation time.” (lines 256-261): “For each of the datasets, the local similarity of each sequence to all the others was calculated. This number is 580*581for Gold Standard Enzymes dataset. Thus, for all datasets, more than 657,398 local similarity calculations were performed using Hewlett-Packard HP X210 Workstation X64-based PC Intel Xeon CPU E31239 3.20 GHz, 8 GB memory for about 1.5 hours, or less than 10 milliseconds per pair of amino acid sequences.”

Point 4. “Consequently, for example, table 3 shows that TMF method is better on 3 cases than the methodology described in the paper. There is just a simple sentence in the conclusion, not supported by any part of the article. Details have to be given to better explain the interest of this new methodology and what does it brings compared to the other one.”

Response 4. We highlighted features of our approaches concerning the study of the diverge protein families (lines 279-285): “As the specificity of ligand-binding can be related with a few number of residues [31, 34], estimation of impact of the individual residues is better suited in the case of the diverged protein family. This situation is typical and can be illustrated by the example of protein kinases. We compiled and verified the sets derived from individual experimental studies on kinase-inhibitor interactions. This study demonstrated reasonable accuracy being tested on the diverse data; thus, our approach significantly extends the applicability domain in proteochemometrics.”

We corrected the English language and style.

Reviewer 2 Report

This manuscript “Prediction of protein-ligand interaction based on the positional similarity scores derived from amino acid sequences” by Karasev et al. proposes a method to predict whether a ligand interact with a target protein based on a score of sequence similarity between the query protein and other proteins in the training data. Basically, these methods need to compare protein sequences with other somehow ‘similar’ proteins. That is why the authors need to collect similar proteins and divide them into specific categories such as enzymes, receptors or ion channels. However, the global sequence similarity does not guarantee the ligand binding, as the local sequence or structural similarity decides the binding instead. Therefore, the authors applied and examined this concept with their ‘novel’ method. Unfortunately, their prediction outcome is not significantly better than the previous ones’. That diminishes the value of this study. Moreover, I have some suggestions on the revision:   

Major issues:

Tables 1 and 4 are exactly the same except titles. The method is not described clearly. There are many equations and variables listed, but some of them are not even well defined or explained, for example, the R in eq. 1. Many equations are mentioned but not explained. It is hard for a reader to understand the rationale, i.e. why this method/equation works. The identification of functionally important residues is more significant than determining whether a ligand bind to a target protein. The authors should demonstrate their method can achieve this goal or find some other ways to increase the overall significance.

Minor issues:

F has been mentioned several times in the results section before it is explained in the method. It is hard for a reader to understand it. It is not explained why the values 7 and 30 were chosen.

Author Response

Dear Reviewer,

We would like to express our gratitude for all the valuable comments and suggestions.

We have tried to do our best to improve our manuscript. Our responses are given below with the new sentences included in text of the manuscript. The modified text of the article in the attached file. The corrections are highlighted by yellow.

Yours sincerely,

On behalf of the co-authors:

Dmitry Karasev

Point 1. “Tables 1 and 4 are exactly the same except titles.”

Response 1. We eliminated this error.  Now Table 4 (lines 167-168) contains the characteristics of datasets.

Point 2. “The method is not described clearly. There are many equations and variables listed, but some of them are not even well defined or explained, for example, the R in eq. 1. Many equations are mentioned but not explained. It is hard for a reader to understand the rationale, i.e. why this method/equation works.”

Response 2. The section “4.2 Positional similarity scores” and “4.3 Prediction algorithm” is rewritten and clarified (lines 203-234).

Point 3. “The identification of functionally important residues is more significant than determining whether a ligand bind to a target protein. The authors should demonstrate their method can achieve this goal or find some other ways to increase the overall significance.”

Response 3. We highlight features of our approaches concerning the study of diverge protein families (lines 279-285): “As the specificity of ligand-binding can be related with a few number of residues [31, 34], estimation of impact of the individual residues is better suited in the case of the diverged protein family. This situation is typical and can be illustrated by the example of protein kinases. We compiled and verified the sets derived from individual experimental studies on kinase-inhibitor interactions. This study demonstrated reasonable accuracy being tested on the diverse data; thus, our approach significantly extends the applicability domain in proteochemometrics.”

Point 4. “F has been mentioned several times in the results section before it is explained in the method. It is hard for a reader to understand it.”

Response 4. We added brief description in the “Introduction” section (lines 59-64) “We developed the method based on the local similarity of amino acid sequences. The sequence fragments of the given length (the F parameter) from the query protein sequence and training protein sequences are compared. By this way, each position of the query sequence gets the score. These values are input data to the classifier, which estimates the protein specificity to the ligands. Earlier, we have demonstrated that this tool can be applied for functional annotation of proteins [37] and identification of functionally important residues in diverged paralog proteins [34].” and in “2. Results and discussion” (lines 72-73): We evaluated the performance of our approach with the length of compared sequence regions (F parameter) equal to 7 or 30.”

We rewritten and clarified the section “4.2 Positional similarity scores” (lines 203-223).

Point 5. “It is not explained why the values 7 and 30 were chosen.”

Response 5. The choice of the F values is explained in a new paragraph in “4.2 Positional similarity scores” (lines 219-223): “The F values were set to 7 and 30 for the following reasons. Based on our preliminary studies, we suggested that the less fragment length is preferable for estimating the impact of the individual amino acid positions [34]. The larger length is better allowed accounting for distant inter-positional dependencies [36].”

We corrected the English language and style.

Round 2

Reviewer 1 Report

Authors have greatly improved the quality of their paper and make the study more clearer.

However, I would have appreciate some details on the comparison with other methods on the special case of protein kinase familiy. I just have few questions : Is it possible for example to apply TMF or CMF method on this dataset? What are the results of those methods ? Is this approach outperforms on this case the existing one ?

I think that adding those details in the paper will make it more impactful. After that, the paper will be suitable for publication in IJMS

Author Response

Dear reviewer,

You pointed out the critical item. Unfortunately, the developers of considered methods had not provided access to their software online or desktop. Therefore, we were not able to compare the efficiency of our approach and others, such as CMF and TMF, related to all considered data sets, including protein kinases and PASS Targets. With your permission, we could not mention it in the paper.

Sincerely yours,

Dmitry Karasev

Reviewer 2 Report

I appreciate the efforts of revising this manuscript.